# EEG Correlates of Active Stopping and Preparation for Stopping in Chronic Tic Disorder

**DOI:** 10.3390/brainsci12020151

**Published:** 2022-01-24

**Authors:** Alonso Zea Vera, Ernest V. Pedapati, Travis R. Larsh, Kevin Kohmescher, Makoto Miyakoshi, David A. Huddleston, Hannah S. Jackson, Donald L. Gilbert, Paul S. Horn, Steve W. Wu

**Affiliations:** 1Department of Neurology, Children’s National Hospital, Washington, DC 20010, USA; 2Department of Pediatrics, George Washington University School of Medicine and Health Sciences, Washington DC 20052, USA; 3Division of Child and Adolescent Psychiatry, Cincinnati Children’s Hospital Medical Center, Cincinnati, OH 45229, USA; ernest.pedapati@cchmc.org; 4Department of Psychiatry, University of Cincinnati College of Medicine, Cincinnati, OH 45267, USA; 5Division of Neurology, Cincinnati Children’s Hospital Medical Center, Cincinnati, OH 45229, USA; Travis.Larsh@cchmc.org (T.R.L.); david.huddleston@cchmc.org (D.A.H.); hannah.jackson@cchmc.org (H.S.J.); Donald.Gilbert@cchmc.org (D.L.G.); Paul.Horn@cchmc.org (P.S.H.); Steve.Wu@cchmc.org (S.W.W.); 6College of Engineering and Applied Science, University of Cincinnati, Cincinnati, OH 45267, USA; kohmeskm@mail.uc.edu; 7Swartz Center for Computational Neuroscience, Institute for Neural Computation, University of California San Diego, La Jolla, CA 92093, USA; mmiyakoshi@ucsd.edu; 8Department of Pediatrics, University of Cincinnati College of Medicine, Cincinnati, OH 45267, USA

**Keywords:** Tourette Syndrome, electroencephalography, stop signal task

## Abstract

Motor inhibition is an important cognitive process involved in tic suppression. As the right frontal lobe contains important inhibitory network nodes, we characterized right superior, middle, and inferior frontal gyral (RSFG, RMFG, RIFG) event-related oscillations during motor inhibition in youth with chronic tic disorders (CTD) versus controls. Fourteen children with CTD and 13 controls (10–17 years old) completed an anticipated-response stop signal task while dense-array electroencephalography was recorded. Between-group differences in spectral power changes (3–50 Hz) were explored after source localization and multiple comparisons correction. Two epochs within the stop signal task were studied: (1) preparatory phase early in the trial before motor execution/inhibition and (2) active inhibition phase after stop signal presentation. Correlation analyses between electrophysiologic data and clinical rating scales for tic, obsessive-compulsive symptoms, and inattention/hyperactivity were performed. There were no behavioral or electrophysiological differences during active stopping. During stop preparation, CTD participants showed greater event-related desynchronization (ERD) in the RSFG (γ-band), RMFG (β, γ-bands), and RIFG (θ, α, β, γ-bands). Higher RSFG γ-ERD correlated with lower tic severity (r = 0.66, *p* = 0.04). Our findings suggest RSFG γ-ERD may represent a mechanism that allows CTD patients to keep tics under control and achieve behavioral performance similar to peers.

## 1. Introduction

Tourette Syndrome (TS) is characterized by persistent motor and vocal tics and affects nearly 1% of the population [1]. TS and the associated co-occurring psychiatric conditions can significantly affect quality of life and result in permanent or life-threatening injuries in severe cases [2,3]. Tics typically start in early childhood, worsen during adolescence, and may decrease by adulthood [4]. The ability to suppress tics, albeit only for brief moments, is a motor control skill that increases with age [5]. It has been postulated that repeated efforts to suppress tics may result in a compensatory enhancement of inhibitory circuits that is critical for the reduction of symptoms over time [6]. Consistent with this theory, a recent study in children with new-onset tics showed that better tic suppression is a predictor of less severe tics one year later [7].

Inhibitory control has been extensively studied in TS using various behavioral paradigms [8]. The interpretation of these studies is confounded by the use of different behavioral tasks [8,9]. Among these approaches, the stop signal task (SST) has the advantage that the experimenter has more stringent control over when the Go and Stop processes begin and allows for the calculation of the latency of the stop process—stop signal reaction time (SSRT) [10,11]. Different models of volitional inhibitory control include reactive inhibition (the outright stopping of an action in response to a signal), and proactive inhibition (the preparation for the possibility of stopping an action) [11]. Most SST studies in TS have focused on reactive inhibition using choice-response SST, where participants give speeded responses to a go signal and attempt to prevent this response when a stop signal is presented. Alternatively, in anticipated-response SST, participants respond when a moving indicator reaches a stationary target while this response is inhibited when a stop signal is presented shortly before the indicator reaches the target [12]. A recent study with healthy adults suggests that anticipated-response SST quantifies SSRT more reliably than choice-response SST [12].

The anatomic localization of inhibitory control has been studied using numerous imaging and physiologic techniques. Multiple functional magnetic resonance imaging (fMRI) studies suggest that the critical network for inhibitory control lateralizes to the right hemisphere and involves the right inferior frontal gyrus (RIFG), right middle frontal gyrus (RMFG), and pre-supplementary motor area (preSMA) [11,13,14]. Functional MRI studies using a modified version of the choice-response SST in healthy adults showed significant activation of the preSMA and RIFG when preparing to stop and outright stopping, suggesting that similar networks are involved in both reactive and proactive inhibition [15,16]. Additionally, a recent meta-analysis found that the RMFG is activated primarily in reactive inhibition and RIFG is recruited mainly for proactive inhibition, suggesting contiguous but separate networks are involved in these processes [13].

While imaging studies have revealed important neuroanatomic regions for inhibitory motor control, the higher temporal resolution of electroencephalography (EEG) and magnetoencephalography (MEG) has demonstrated the underlying physiologic changes [17,18]. In reactive inhibition tasks, an increase in prefrontal β-power has been reported consistently [18,19]. Meanwhile, in proactive inhibition tasks, multiple studies support the role of mid-frontal θ-oscillations [20,21], as well as δ and high-γ oscillations [15,20].

Functional MRI has often been used to examine frontal circuits believed to underlie compensatory adaptation in tic disorder patients. For example, TS participants exhibit higher frontal BOLD signals during blink suppression compared to controls and during voluntary tic suppression [22,23,24]. Furthermore, overlapping frontal regions are over-activated during blink inhibition and cognitive control tasks in TS patients, suggesting an exaggerated deployment of behavioral inhibition circuits that extend beyond purely motor and into the cognitive domain [22,25]. Other fMRI studies have also found differences in frontal activity during behavioral inhibition tasks between TS patients and controls [9,26,27,28], with some studies identifying significant correlations between tic severity and frontal BOLD activity [25,26].

MEG and EEG physiology studies in TS have focused on the sensorimotor area. TS patients show abnormalities in movement-related β-band desynchronization, a well-known movement-related oscillation, contralateral to hand movements [29,30,31]. Interestingly, active tic suppression appears to normalize this movement-related cortical activity [30]. Furthermore, compared to healthy controls (HC), TS patients have been found to have increased coherence between prefrontal and sensorimotor areas during motor inhibition and tic suppression [29,32,33].

This study aimed to evaluate inhibitory control in chronic tic disorders (CTD) by comparing changes in the cortical electrophysiology in frontal areas critical for motor inhibition and exploring the relationship of these physiologic findings to clinical measures. We used an anticipated-response SST, where the stop signal precedes the anticipated go response [12]. Using dense array EEG allowed us to examine physiologic changes at high temporal resolution, while EEG source localization technique improved the spatial resolution of our analysis. We hypothesized that activation of frontal regions during the anticipated-response SST would differ in CTD versus HC and would correlate with clinical symptoms.

## 2. Materials and Methods

### 2.1. Participants

Children between 10–17 years old who fulfilled DSM-5 diagnostic criteria for TS or another CTD were recruited from the Cincinnati Children’s Hospital Medical Center (CCHMC) Movement Disorders Clinic. Comorbid neuropsychiatric conditions and stable doses of psychotropic medications were allowed. Typically developing HC were recruited through the Institutional Review Board (IRB)-approved flyers and online advertisements.

Clinical symptoms were assessed using validated scales—Yale Global Tic Severity Scale (YGTSS) Total Tic Scale [34], Premonitory Urge for Tic Scale (PUTS) [35], Children’s Yale–Brown Obsessive–Compulsive Scale (CY-BOCS) [36], and DuPaul ADHD Rating Scale-IV [37]. PUTS is a 10-question self-reported scale with the first nine questions assessing sensory experience relating to the patients’ tics. The last question asks how well the patients can suppress their tics and the value for this question was used to represent “tic suppressibility” in statistical analyses. Written informed consent was obtained from the parent or legal guardian of study participants. Children also gave written assent. The study was approved by the CCHMC IRB (IRB protocol number 20081636).

### 2.2. Anticipated Response Stop Signal Task (Slater-Hammel Task)

We used a child-friendly version of the anticipated-response SST that we developed to assess behavioral motor inhibition (Figure 1) [38]. A video showing the task is available at: https://www.jove.com/v/56789 (accessed date: 21 January 2022). The task was presented on a computer monitor using Presentation^®^ software (v. 10.0; Neurobehavioral Systems, Albany, CA, USA) while participants were seated in a comfortable chair. The ulnar aspects of both arms and hands rested on a body-surrounding pillow while the palmar surface faced medially. Subjects used their dominant thumb to operate a game controller to complete this task. Details of the task are presented in Figure 1.

The go:stop trial ratio was 2:1. The initial stop signal delay (SSD) for each of the blocks occurred at 500 ms, then shifted by 50 ms using a staircase procedure depending on the success of stop trials to reach an average probability of successfully stopping of ~0.5 (Figure 1C). Participants practiced with 10 only go trials, 10 only stop trials, and 20 mixed go and stop trials. If the study team judged that the child understood the instructions, the game was played with four 40-trial mixed blocks for a total of 160 trials.

We calculated the percentage of successful go trials (defined as stopping between 700–800 ms), average SSD, average go trial finger lift time (go reaction time; GO-RT), and stop signal reaction time (SSRT) for each block. Since we used a staircase procedure to achieve ~0.5 success rate in all participants, SSRT for each block was calculated using the “means” method described in the consensus guide by Verbruggen et al. [10]. Therefore, the SSRT is the result of subtracting the average SSD from the average GO-RT.

### 2.3. EEG Recording and Pre-Processing

EEG data were recorded using an EGI NetAmp 300 system with a 128-channel electrode cap (MagstimEGI, Eugene OR) at 1000 Hz sampling rate. The interface between Presentation^®^ software and EEG acquisition software (Net Station) occurred through the TTL functionality such that SST events were marked on the EEG tracing.

EEG data analysis was performed using custom scripts in MATLAB R2018a (MathWorks, Natick, MA, USA), EEGLAB (v2019.1), and Brainstorm (version 83) [39,40]. The code is available upon request. Initially, EEG data were high-pass filtered at 2 Hz and a 60 Hz notch filter was applied. Next, EEG tracing was visually inspected for artifact removal. Further removal was performed by identifying channels that correlated less than 0.6 with neighboring channels using EEGLAB clean-rawdata plugin. Similarly, trials with large artifacts were first removed using visual inspection, followed by an automated process using EEGLAB’s trial rejection GUI based on data statistics. First, trials with high amplitudes that exceeded +/−300 µV were rejected. This was followed by rejecting “improbable” epoched data based on channels exceeding 4 standard deviations using EEGLAB. Given greater proportion of EEG artifacts in the CTD group (see Results), we used these criteria to avoid excess trial rejection [41].

Subsequently, removed EEG channels were interpolated and were re-referenced to a common average reference. Independent component analysis was then performed using binica.m (extended infomax) function in EEGLAB, followed by removal of components consistent with eye blinks/saccades and heartbeat artifacts.

### 2.4. EEG Source Modeling

To address the inverse problem, we performed distributed source modeling in Brainstorm [40] so that time-frequency analysis was carried out at the source rather than at the electrode sensor level. First, individual T1 images were segmented using the Computational Anatomy Toolbox for SPM12. Most subjects (HC—9/13; CTD—10/14) had their own MRI images. For those who had dental hardware or whose MRI images were degraded due to motion artifacts, age-based templates [42] were used. We then performed source estimation through the following steps. Forward modeling was achieved by the Symmetric Boundary Element Method using the open-source software OpenMEEG. This model uses three realistic layers with 1922 vertices per layer (default setting). The minimum norm estimate (MNE) imaging method was used to solve the inverse problem. Dipole orientations at each grid point were constrained to a normal orientation in relation to the cortex surface. Finally, time series data were extracted based on the Desikan–Killiany atlas [43] and exported for time-frequency analysis using custom Matlab codes. Extracted regions represented the current source density averaged across vertices representing the Desikan–Killiany regions of interest (ROI). We focused on three right frontal regions: right superior frontal gyrus (RSFG), which includes pre-SMA, RMFG (averaged from the Desikan–Killiany right rostral middle frontal and caudal middle frontal ROI) and RIFG (averaged from the Desikan–Killiany right pars orbitalis, pars triangularis, and pars opercularis ROI), as these are critical for inhibitory control [11,13].

### 2.5. EEG Analysis

Time-frequency decomposition of the extracted time series from the ROIs was achieved using complex Morlet wavelet convolution [44]. We used 95 wavelets with frequencies linearly increasing from 3 to 50 Hz in 0.5 Hz increments. Power decibel (dB) baseline normalization was performed using the last 200 ms of the fixation screen (between “READY” and “GET SET”; Figure 1A) as the baseline period. To assess outright stopping, we analyzed all stop trials and the presentation of the stop signal (Figure 1A) was used as time latency zero. To assess the preparation of the stopping process we analyzed all go and stop trials and the initiation of car motion (Figure 1A) was used as time latency zero. The period of interest for this second analysis was between the start of car moving (time latency 0 ms) and 300 ms. This window was used because the stop signal in our task can appear anytime from 300–700 ms (Figure 1A). Therefore, before 300 ms, participants are unaware of the trial type. During this epoch, participants are preparing for possibly stopping but are uncertain if they will need to inhibit their responses. After 300 ms, some trials might be contaminated by the presence of stop signals and active stopping. 

Group comparison was performed first by combining dB-converted data into a single master array (frequency × time × trials). An array representing group differences was calculated by averaging all the trials within each group and subtracting the group averages. Next, a permutation array was generated by permuting the trial dimension of the master array (1000 times) and calculating the difference between group averages. Multiple comparison correction was performed using the cluster-based method. First, the permutation and group-difference arrays were converted to both Z and t-statistic values. The permutation array was then thresholded at *p* = 0.05. After this initial uncorrected thresholding, surviving time-frequency clusters were identified using the bwconncomp function from the Matlab Image Processing Toolbox. Within each cluster, the sum of the absolute t-statistic values was calculated for each of the 1000 permutations to create a distribution of summed (absolute) t-statistics. The absolute t-statistic sum threshold for multiple comparison correction was identified as the 98.3rd percentile of this distribution (i.e., using *p* = 0.017 to account for multiple comparisons of 3 right frontal regions). Clusters above this 98.3rd percentile threshold were considered statistically significant.

### 2.6. Statistical Analysis

All statistical analyses were performed using SAS^®^ (SAS Institute, Inc., Cary, NC, USA). Data normality was tested using the Shapiro–Wilk test. Data are presented as mean ± standard deviation. Demographic and EEG quality control data were compared between diagnostic groups using the Wilcoxon–Mann–Whitney test (for age, number of rejected channels/independent components), two-sample t-test (for number of rejected trials), or Fisher’s exact test (for sex) as appropriate. All tests were two-tailed with α < 0.05 used to define statistical significance.

We performed a repeated measures mixed model analysis using the GLIMMIX procedure in SAS^®^ to analyze behavioral data. SST behavioral measures (percentage of successful go trials, GO-RT, SSD, and SSRT) were analyzed as dependent variables. Diagnosis, block, and diagnosis*block interaction were included in the model as independent variables. Age was included as a covariate. Diagnosis*block interaction was the primary interaction of interest.

We performed exploratory Pearson correlation analyses to evaluate the relationship between EEG power dB change in the time periods of interest and clinical measures (YGTSS, CY-BOCS, and DuPaul ADHD scale scores). Age-adjusted correlation analyses were performed, as tic severity can vary throughout course of childhood and adolescence [4]. This analysis was only performed for statistically significant case–control differences in EEG power dB change. We decided a priori to analyze each ROI separately. Based on the case–control time-frequency difference found, we generated a mask to extract the power-dB averages. The age-adjusted correlation between the extracted EEG dB power change and YGTSS, CY-BOCS, and DuPaul ADHD scale scores was assessed, followed by multiple comparison correction using the false discovery rate (FDR) with a family-wise error rate = 0.05 [45].

## 3. Results

### 3.1. Demographic, Clinical, and Behavioral Data

All 27 participants were right-handed except one TS participant. Age difference was not significant (*p* = 0.36) between HC (*n* = 13; 12.7 +/− 2.8 years) and CTD (*n* = 14; 13.3 +/− 1.9 years) groups. The difference in sex proportion was not significant between groups—six HC females and three CTD females (*p* = 0.24). Clinical characteristics of the CTD participants are presented in Table 1. The mean total YGTSS tic score was 23 (SD 9.3) with a median of 23 and the mean total PUTS score was 22 (SD 5.9) with a median of 20. We found no significant correlation between the YGTSS total tic score and PUTS score (unadjusted *p* = 0.1). 

SST performance was similar between CTD and HC groups (Table 2). The primary variable of interest for the behavioral data was the diagnosis*block interaction in the repeated measures mixed model analysis. The diagnosis*block interaction was not significant in all models using behavioral measurements (probability of inhibiting, percentage of successful go trials, GO-RT, SSD, and SSRT) as dependent variables (all *p* > 0.05). CTD participants who had faster SSRT had self-reported better ability at suppressing their tics (r = −0.58, *p* = 0.03). No significant correlation was found between SSRT and other clinical measures. 

### 3.2. EEG: Quality Control 

No between-group differences were found in number of rejected channels (*p* = 0.98) and rejected independent components (*p* = 0.46). More trials (*p* = 0.02) were rejected due to artifacts in the CTD group (mean 36.5%, SD 13.6%) than controls (mean 23.5%, SD 12.5%). 

### 3.3. EEG: Outright Stopping (Stop Trials)

During successful stop trials, we found event-related synchronization (ERS) spanning all the analyzed frequencies (3–50 Hz) in the RSFG, RMFG, and RIFG (Appendix A) for HC and CTD subjects. However, the most prominent ERS occurred in the θ and α frequency bands. After adjusting for multiple comparisons, we found no statistical difference in the time-frequency domain between HC and CTD, or between successful vs failed stop trials in either group.

### 3.4. EEG: Preparation for Possibly Stopping (All Trials)

In the RSFG (Figure 2), RMFG (Figure 3), and RIFG (Figure 4), CTD and HC exhibited significant γ event-related desynchronization (ERD) starting slightly before the car started moving (time latency 0). However, CTD patients showed significantly greater γ-ERD than HC. In the RMFG (Figure 3), group ERS differences were also found in the high β band, whereas in the RIFG (Figure 4), additional group ERD differences were observed in θ, α and β bands. In CTD participants, γ-ERD in the RSFG region significantly correlated to lower tic severity (Figure 5; r = 0.66, FDR adjusted *p* = 0.04). In other words, CTD subjects who had greater γ-ERD during the beginning of the trial (0–300 ms) had lower YGTSS total tic severity scores.

## 4. Discussion

We used an anticipated-response SST and ROI EEG source localization approach to investigate right frontal cortical physiology associated with motor control in children with CTD. SSRT was similar between HC and CTD patients, consistent with prior reports [8,46]. For outright stopping, the right frontal activation pattern did not differ between groups. However, during preparation for stopping, CTD subjects exhibited higher γ-ERD in the RSFG, RMFG, and RIFG. During this evaluated period, there was uncertainty about the trial type and thus participants were preparing for the possibility of stopping. Furthermore, more extensive RSFG γ-ERD correlated with lower tic severity in CTD patients, but not measures of ADHD or obsessive-compulsive disorder. The findings in this small cohort of pediatric patients with chronic tics suggest that this right prefrontal γ-ERD may represent a mechanism used to keep tics under control and allow CTD patients to achieve similar SST performance as peers.

Increased RSFG γ-ERD during an SST in TS/CTD has not been reported previously. However, we could not find EEG studies evaluating a similar behavioral task and ROIs. Functional MRI studies using an anticipated-response SST have shown that the RSFG is an important region during task performance. In healthy adults, increased activation of the supplementary motor complex is observed before stop signal cue, potentially representing the expectation of the stop signal [47]. Physiologically, prefrontal γ activity has been reported to increase during cognitive tasks in healthy adults [48,49,50] and during cued blinking in TS patients [51]. In animal studies, β and γ (20–40 Hz) changes in the supplementary motor complex have been detected when updating a motor plan [52]. Since motor inhibition is a subconstruct of motor control, prefrontal γ activity may represent physiologic mechanisms that underlie motor planning when deciding between motor execution vs inhibition [53,54].

One other possible interpretation of our findings is that the increased γ-ERD represents prefrontal activity involved in motor control in CTD patients. This γ-ERD was seen before the presentation of the stop signal in regions where both reactive and proactive inhibition have been observed [13]. Different inhibition models have been proposed and examined in tic patients [55], with volitional inhibition (i.e., reactive and proactive) most commonly studied. However, results have been mixed. Phenomenologically, tic suppression could involve both reactive (e.g., suppressing when instructed) and proactive (e.g., preparing to suppress tics depending on different situations/settings) inhibition. Results for both reactive and proactive inhibitory deficit in CTD have been mixed [8]. Other forms of inhibition (e.g., automatic inhibition) has also been shown to be inconsistent in the tic population [56,57]. Consistent with multiple published studies, our results demonstrated similar volitional reactive inhibition between controls and CTD. However, differences in right frontal spectral activities were found during time windows when one needed to decide whether to execute motor action vs inhibit it. Although this may be somewhat analogous to proactive inhibition, a more complex task design would be necessary to specifically address the underlying physiologic activities (see limitation paragraph below).

Alternatively, our findings could be explained by γ oscillation’s prominent role in the default mode network (DMN) [58]. The DMN includes several cortical nodes, including prefrontal areas, that are functionally connected and active during resting wakefulness. Changes in DMN activity depend on the nature of the directed activity at hand, with externally triggered tasks resulting in decreased overall activity [59]. Resting-state fMRI studies have shown functionally immature brain networks in teenagers with TS and hyperconnectivity of the DMN in adults with TS [60,61]. Furthermore, increased DMN connectivity correlated with lower tic severity, suggesting a role as a compensatory mechanism for tic suppression [60]. Since DMN activity decreases with externally cued tasks, such as the SST, we speculate that TS participants may need to mount a more robust γ-ERD due to their hyperconnected DMN. Consistent with this idea, we found that lower tic severity correlated with higher γ-ERD. An extension of this study by comparing resting-state vs task-related signals may help to explore these hypotheses.

Our study is the first to assess EEG activity during an anticipated-response SST in CTD. Although we did not find electrophysiological differences between HC and CTD during outright stopping, we observed ERS in the right frontal regions in both groups. This finding is consistent with prior adult motor inhibition studies, with much attention focused on β oscillatory activity [18,19,62], and supports the validity of our motor inhibition paradigm. Further exploration of this paradigm may provide additional insights into tic pathophysiology and suppression. However, additional studies are needed to define EEG signatures specific to anticipated-response SST in healthy subjects and chronic tic patients.

There are several limitations to this study. First, our SST was self-paced, which allowed time during the task for CTD children to tic and get ready for the next trial. However, this results in different inter-trial intervals that could affect performance in motor inhibition tasks [63]. Second, we used a high percentage (33%) of stop trials, which could have affected our capacity to detect motor inhibition differences between groups by inducing a strategic slowing of responses by the participants [10]. This was partially mitigated by using an anticipated-response SST and including feedback during the task. Third, we share the limitations inherent to scalp EEG studies and the inverse problem associated with source estimation. When available, we ameliorated this issue by using each individual’s brain MRI for source localization so we could analyze activity from similar regions of the brain across the cohort. Fourth, we did not monitor for tics therefore we cannot assess how these could have affected the EEG signals. Fifth, we did not include variable probabilities of stop trials, which is necessary to assess proactive inhibition [47]. Finally, our small sample size did not allow us to evaluate the effect of psychiatric comorbidities and psychotropic medications that could confound our results. Consequently, we only adjusted for age in our exploratory correlation analyses as the small sample does not support adjusting for multiple confounding factors. Our sample size could also result in missed behavioral differences between groups. Despite these limitations, we were able to find a significant electrophysiologic signature in TS that correlated with disease severity. In the future, a larger study would be needed to address the limitations of our study.

## 5. Conclusions

In summary, we used dense array EEG and a child-friendly anticipated-response SST to examine frontal brain oscillations associated with motor inhibition. We found that CTD patients showed increased γ-ERD in the right prefrontal regions during preparation for stopping, but before the decision to stop was made. Furthermore, this higher RSFG γ-ERD was associated with lower tic severity, suggesting it could reflect a mechanism for tic inhibition. Multiple future avenues can be taken to further understand this finding in TS, such as comparing γ activity between rest vs task performance and using non-invasive brain stimulation to modulate RSFG γ oscillations and assess its effect on motor inhibition and tic control.

## Figures and Tables

**Figure 1 brainsci-12-00151-f001:**
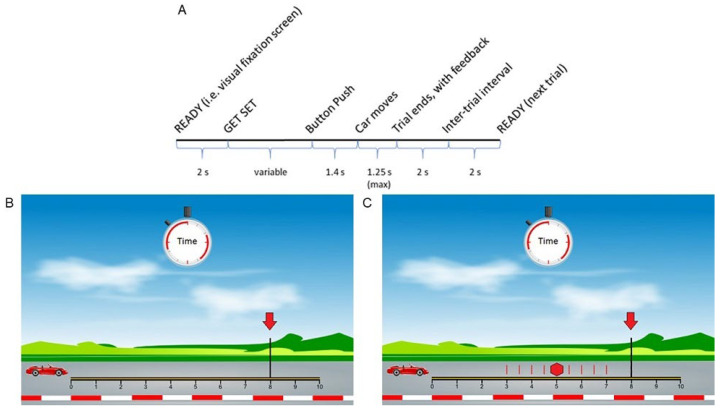
Anticipated-response stop signal task. (**A**) Timeline of one trial of the stop signal task. An initial fixation screen with the word “READY” in the middle of the screen was shown for 2000 ms. After the “GET SET” cue, participants were instructed to push and hold the game controller button in a self-paced manner to begin the trial. After 1400 ms, the car moved for 1000 ms across the racetrack on the screen. (**B**) Go trial. The car kept moving only as long as the button remained pressed. Participants were instructed to release the button after 700 ms and as close to the 800 ms mark (red arrow), but without going past it. (**C**) Stop trial. In randomly interspersed trials, the car stops spontaneously (i.e., stop cue, illustrated by red stop sign) between 300–700 ms (red vertical lines). Participants were instructed to keep the button pressed until a checkered flag appears on the screen (at 1000 ms). Feedback was provided for each trial. For go trials: “Too early!” for action at <700 ms; “Great job!” for action from 700 to 800 ms; “Too late!” for action at >800 ms. For stop trials: “Too early!” for button hold < 1000 ms after stop cue (failed stop); “Great job!” for button hold > 1000 ms (successful stop).

**Figure 2 brainsci-12-00151-f002:**
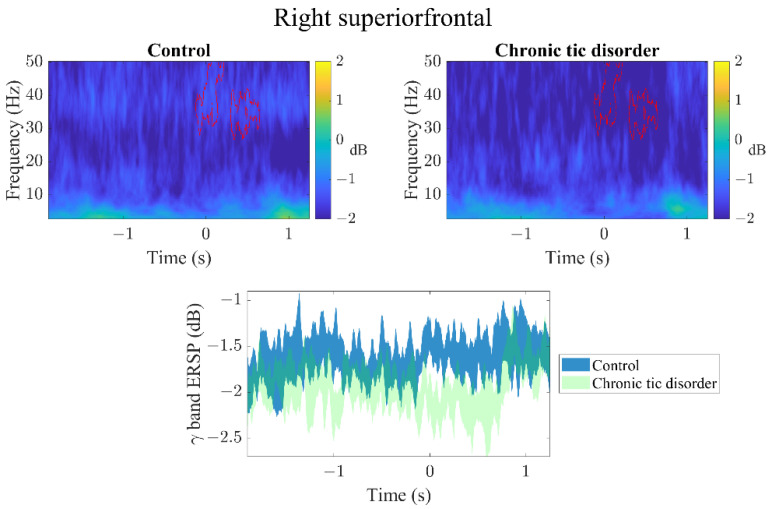
All (GO + STOP) trial event-related spectral perturbation images of the right superior frontal region. Latency time 0 represents when the car starts moving. Upper row: Both groups show γ event-related desynchronization (ERD) compared to baseline beginning slightly before the car started moving. Chronic tic disorder participants showed significantly greater γ ERD (represented by red contour area). Lower row: Average γ-band (30–50 Hz) ERD change over time. Around the time when the car started moving (latency time 0), the chronic tic disorder group showed greater γ-band ERD.

**Figure 3 brainsci-12-00151-f003:**
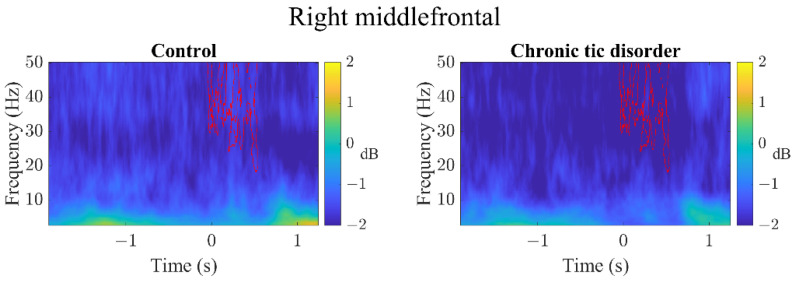
All (GO + STOP) trial event-related spectral perturbation images of the right middle frontal region. Latency time 0 represents when the car starts moving. Both groups show event-related desynchronization (ERD) beginning slightly before the car started moving. The chronic tic disorder group showed significantly greater ERD across in β and γ bands (represented by red contour area).

**Figure 4 brainsci-12-00151-f004:**
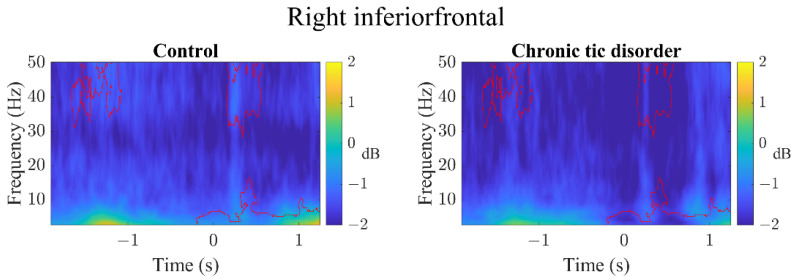
All (GO + STOP) trial event-related spectral perturbation images of the right inferior frontal region. Latency time 0 represents when the car starts moving. Both groups show event-related desynchronization (ERD) beginning slightly before the car started moving. The chronic tic disorder group showed significantly greater ERD across multiple frequency bands (represented by red contour area).

**Figure 5 brainsci-12-00151-f005:**
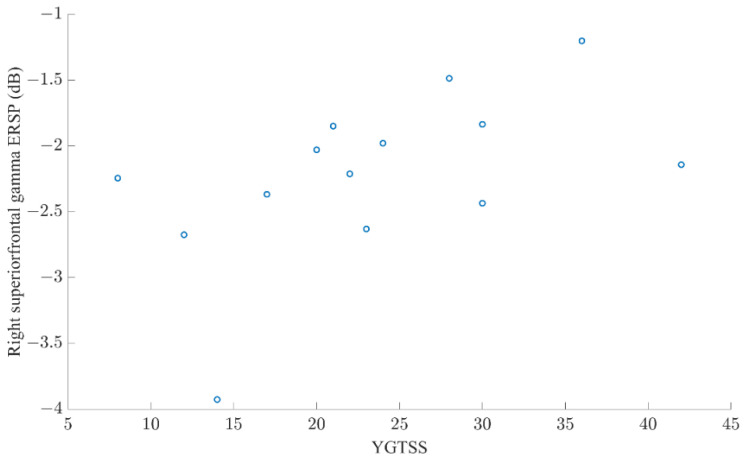
Scatter plot of right superior frontal γ event-related spectral perturbation (ERSP) and Yale Global Tic Severity Score total tic score. Subjects with lower tic severity showed greater γ event-related desynchronization (r = 0.66, FDR adjusted *p* = 0.04).

**Table 1 brainsci-12-00151-t001:** Demographic Characteristics.

Participant	Diagnosis	Age	Sex	Handed-ness	YGTSS	PUTS	DuPaul ADHD Scale	CY–BOCS	Medication(s)
1	TS	13	M	L	30	19	21	0	citalopram, gabapentin
2	TS	14	F	R	22	11	15	1	none
3	TS	11	M	R	8	18	5	0	none
4	TS	14	M	R	21	26	11	0	none
5	TS	11	M	R	17	19	13	0	none
6	TS	12	M	R	36	16	18	16	citalopram, clonidine, risperidone
7	TS	15	M	R	42	21	40	21	atomoxetine, fluvoxamine
8	TS	14	M	R	23	29	21	0	guanfacine, sertraline
9	TS	11	M	R	20	23	22	0	none
10	CMTD	11	F	R	12	19	4	0	none
11	TS	16	M	R	14	28	29	22	desvenlafaxine
12	TS	15	F	R	30	31	30	22	none
13	TS	16	M	R	24	15	0	18	none
14	TS	13	M	R	28	27	13	0	None

CMTD = chronic motor tic disorder, CY–BOCS = Children’s Yale–Brown Obsessive–Compulsive Scale, ADHD = Attention-Deficit/Hyperactivity Disorder, TS = Tourette Syndrome, PUTS = Premonitory Urge for Tics Scale, YGTSS = Yale Global Tic Severity Scale total tic score.

**Table 2 brainsci-12-00151-t002:** Performance in anticipated-response Stop Signal Task.

	Chronic Tic Disorders (*n* = 14)	Healthy Controls (*n* = 13)	*p* Value (Diagnosis * Block)
Probability of Inhibiting	0.54 ± 0.04	0.54 ± 0.06	0.54
Stop Signal Reaction Time (ms)	249.6 + 36.3	249.4 + 40.5	0.70
Stop Signal Delay (ms)	548.4 ± 33.9	543.8 ± 44.8	0.35
Go Reaction Time (ms)	798.0 ± 25.2	793.2 ± 22.6	0.15
Go trial success rate *	0.48 ± 0.15	0.53 ± 0.16	0.44

Mean ± standard deviation shown. * Go trial success was defined as lifting finger within the 700–800 ms time window.

## Data Availability

Unidentified data is available upon reasonable request.

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
