# Peer review of "EEG Correlates of Active Stopping and Preparation for Stopping in Chronic Tic Disorder"

_brainsci, 2022, doi:10.3390/brainsci12020151_

Round 1

Reviewer 1 Report

Authors asked to 14 children with CTD and 13 controls (10-17 years old) to perform an anticipated-response SST while EEG was recorded. Results showed no behavioral or EEG differences for reactive stopping condition. Whereas in preparing to stop condition, CTD participants showed greater ERD in the RSFG (γ-band) and RIFG (θ, α, β, γ -bands). Additionally higher RSFG γ-ERD correlated with lower tic severity (=0.57, p=0.04). Authors speculate that RSFG γ-ERD may represent a compensatory process allowing CTD patients to keep tics under control during preparation for stopping. 
This is a well-designed and interesting study. In my opinion the paper is of potential interest for publication. However, I believe that several sections of the manuscript need to be improved. In my view the manuscript is not ready for publication yet.
These are my comments:
1.- In introduction authors neglected relevant metanalyses and current advancement in the neural correlates of cognitive control:
"Multiple lines of evidence suggest that the critical network for outright stopping is lateralized to the right hemisphere and involves the right inferior frontal gyrus (RIFG) and pre-supplementary motor area (preSMA).[11]"
Authors only cited one study of 2011. I believe that this section must be improved with more recent literature on cognitive control. In particular, Gavazzi et al., (2020) should be at least mentioned, because these authors explored recently with ALE metanalyses the domain of cognitive control, re-runned Simmonds et al., (2008) with the updated ALE algorithm and proposed a model of cognitive control disentangling the neural correlates of proactive and reactive inhibition. In particular, they spatially differentiated areas involved in proactive and reactive components of inhibition. According to authors the inhibitory component involved in proactive processes mainly recruits the r-IFG, whereas the inhibitory component employed in reactive processes mainly engages the rMFG.
Simmonds, D. J., Pekar, J. J., & Mostofsky, S. H.. Meta-analysis of go/no-go tasks demonstrating that fMRI activation associated with response inhibition is task-dependent. Neuropsychologia 2008
Gavazzi, G.; Giovannelli, F.; Currò, T.; Mascalchi, M. Contiguity of proactive and reactive inhibitory brain areas: A cognitive model based on ALE meta-analyses. Brain Imaging Behav. 2020.

2. - 2.3 EEG recording and pre-processing
Authors wrote: "First, trials with high amplitudes that exceeded +/- 300 µV were rejected. This was followed by rejecting “improbable” epoched data based on channels exceeding 4 standard deviations."
Nothing can be done at this point, however, these criteria seem to me pretty generous, even if an ICA is applied to correct the signal alterations due to eye blinks/saccades and heartbeat artifacts.
The authors should better motivate such choices in the paragraph or at least report some methodological paper that sustains their choices.

3. -2.3 EEG recording and pre-processing :
“We focused on two ROI: right superior frontal gyrus (RSFG), which includes pre-SMA, and RIFG (averaged from the Desikan-Killiany right pars orbitalis, pars triangularis, and pars opercularis ROI), as these are critical for inhibitory control.[11]”
Similarly to point 1, the recent progress in the exploration of the inibitory neural network of  shows a crucial involvement of right middle frontal gyrus  (Gavazzi et al., 2020), that here is not considered.
It would enrich the paper to replicate the analysis adding a further ROI based on the right middle frontal gyrus.

4. - Discussion:
I believe that discussion possibly needs to be rearranged depending on the results obtained analyzing the right middle frontal gyrus activity

Author Response
Authors asked to 14 children with CTD and 13 controls (10-17 years old) to perform an anticipated-response SST while EEG was recorded. Results showed no behavioral or EEG differences for reactive stopping condition. Whereas in preparing to stop condition, CTD participants showed greater ERD in the RSFG (γ-band) and RIFG (θ, α, β, γ -bands). Additionally higher RSFG γ-ERD correlated with lower tic severity (=0.57, p=0.04). Authors speculate that RSFG γ-ERD may represent a compensatory process allowing CTD patients to keep tics under control during preparation for stopping. 
This is a well-designed and interesting study. In my opinion the paper is of potential interest for publication. However, I believe that several sections of the manuscript need to be improved. In my view the manuscript is not ready for publication yet.
These are my comments:
1.- In introduction authors neglected relevant metanalyses and current advancement in the neural correlates of cognitive control:
"Multiple lines of evidence suggest that the critical network for outright stopping is lateralized to the right hemisphere and involves the right inferior frontal gyrus (RIFG) and pre-supplementary motor area (preSMA).[11]"
Authors only cited one study of 2011. I believe that this section must be improved with more recent literature on cognitive control. In particular, Gavazzi et al., (2020) should be at least mentioned, because these authors explored recently with ALE metanalyses the domain of cognitive control, re-runned Simmonds et al., (2008) with the updated ALE algorithm and proposed a model of cognitive control disentangling the neural correlates of proactive and reactive inhibition. In particular, they spatially differentiated areas involved in proactive and reactive components of inhibition. According to authors the inhibitory component involved in proactive processes mainly recruits the r-IFG, whereas the inhibitory component employed in reactive processes mainly engages the rMFG.
Simmonds, D. J., Pekar, J. J., & Mostofsky, S. H.. Meta-analysis of go/no-go tasks demonstrating that fMRI activation associated with response inhibition is task-dependent. Neuropsychologia 2008
Gavazzi, G.; Giovannelli, F.; Currò, T.; Mascalchi, M. Contiguity of proactive and reactive inhibitory brain areas: A cognitive model based on ALE meta-analyses. Brain Imaging Behav. 2020.

RESPONSE
We thank the reviewer for this helpful suggestion. We have edited the third Introduction paragraph and also included these two references.
Additionally, we have added an analysis of the RMFG (see the response to comment 3).

  1. - 2.3 EEG recording and pre-processing

Authors wrote: "First, trials with high amplitudes that exceeded +/- 300 µV were rejected. This was followed by rejecting “improbable” epoched data based on channels exceeding 4 standard deviations."
Nothing can be done at this point, however, these criteria seem to me pretty generous, even if an ICA is applied to correct the signal alterations due to eye blinks/saccades and heartbeat artifacts. The authors should better motivate such choices in the paragraph or at least report some methodological paper that sustains their choices.

RESPONSE
We used a combination of automated and manual methods for artifact rejection. Given that many CTD participants had tics involving eyes and facial muscles, many EEG tracings had significant high-amplitude artifacts. In order to preserve as many successful stop trials for analysis, we chose a high amplitude threshold for rejection. For the “improbable” epoch rejection, other papers studying tic patients have used similar criteria (e.g. Morera Maiquez et al. Journal of neuropsychology. 2021 used 5 SD as rejection criteria).

  1. -2.3 EEG recording and pre-processing:

“We focused on two ROI: right superior frontal gyrus (RSFG), which includes pre-SMA, and RIFG (averaged from the Desikan-Killiany right pars orbitalis, pars triangularis, and pars opercularis ROI), as these are critical for inhibitory control.[11]”
Similarly to point 1, the recent progress in the exploration of the inibitory neural network of shows a crucial involvement of right middle frontal gyrus (Gavazzi et al., 2020), that here is not considered. It would enrich the paper to replicate the analysis adding a further ROI based on the right middle frontal gyrus.

RESPONSE
We completed analysis for the right middle frontal region. This data is now included in the revised manuscript.  After multiple comparison correction for three right frontal regions, the right middle frontal region shows difference in beta and gamma bands between controls and tic patients.  However, the EEG activity in the right middle frontal region did not significantly correlate with clinical symptoms.

  1. - Discussion:

I believe that discussion possibly needs to be rearranged depending on the results obtained analyzing the right middle frontal gyrus activity.

RESPONSE
We have made changes to the discussion to incorporate the findings of the right middle frontal gyrus.

Reviewer 2 Report

  1. The authors claim that the increase in gamma desynchronisation may be a compensatory mechanism that allows for normal response inhibition. The term compensation implies that there is a deficit elsewhere that gamma power is compensating for – the absence of such a deficit in this study undermines the claim that gamma ERD is compensatory. Increased gamma desynchronisation might alternatively be explained by the presence of ADHD/OCD or medication status.
  2. The authors have adjusted for age but might also want to adjust for OCD, ADHD and medication use (see above).
  3. It is becoming clearer that voluntary aspects of movement control (preparation, execution and inhibition) are normal in patients who have tics. This current study adds to that growing body of literature. There is a question whether tics are due to inappropriate motor activation (like epileptiform discharges) or due to abnormal motor inhibition. There is evidence that the latter causes tics (see https://doi.org/10.1093/brain/awaa024). I suggest that the authors discuss how their study contributes to how tics are generated vs how they are inhibited.
  4. The authors assess the neurophysiology when stopping might be expected. If I’m not mistaken, this sounds like proactive inhibition (see https://doi.org/10.1038/nrn4038 for a good review). The EEG correlates of proactive inhibition have been extensively studied and some background reading might help the authors frame their finding in the wider context.
  5. There are multiple cognitive processes that occur after the car starts moving. Participants prepare to lift their finger as the car approaches the line and (as the authors state) the expectation that a stop signal appears. The current experimental paradigm does not differentiate which of these two processes causes the EEG changes. The study lacks a key control condition – a block of go trials only, which eliminates one of the possibilities above (the expectation that they may need to stop).
  6. There are several areas in the manuscript where the authors correlate physiological and clinical measures, and state that there are no significant interactions (without giving specifics of the comparisons made). Please state if the comparisons were corrected for multiple comparisons and if so, how many. This is especially important given that their correlation only just reaches statistical significance (p=0.04).
  7. Please include a table of the behavioural results including p(inhibit), SSRT, SSF, reaction times, error rates.
  8. If p(inhibit) deviates from 0.5, then the authors may want to consider using the integration method rather than the mean method for calculation of the SSRT.
  9. Preparation of stopping process: I understand why the authors have looked at the time-period between 0 and 300ms (because the stop signal can appear anytime between 300 and 700ms. I wonder if the authors are missing out on significant information about movement preparation by not extending the timepoint beyond 300ms. Indeed, the mean SSD for each participant will be known and instead, the analysis can be re-performed from 0ms to the mean SSD for each participant.

Author Response

1.- The authors claim that the increase in gamma desynchronisation may be a compensatory mechanism that allows for normal response inhibition. The term compensation implies that there is a deficit elsewhere that gamma power is compensating for – the absence of such a deficit in this study undermines the claim that gamma ERD is compensatory. Increased gamma desynchronisation might alternatively be explained by the presence of ADHD/OCD or medication status.
RESPONSE
We thank the reviewer for this comment. We have removed the term “compensation” to avoid overstating the interpretation of the results. We have modified the text: “Furthermore, more extensive RSFG γ-ERD correlated with lower tic severity in CTD patients, but not measures of ADHD or obsessive-compulsive disorder (OCD). The findings in this small cohort of pediatric patients with chronic tics suggest that this right prefrontal γ-ERD may represent a mechanism used to keep tics under control and allowing CTD patients to achieve similar SST performance as peers.”

2.- The authors have adjusted for age but might also want to adjust for OCD, ADHD and medication use (see above).
RESPONSE
We thank the reviewer for this suggestion.  We repeated partial correlation analysis accounting for medical and comorbidity status. These results were not significant. After discussing with our statistician/co-author (P.S.H), we decided not to report this result as the small sample does not rigorously support including three variables (age, medication status, comorbidity status) in the partial correlation analyses.

3.- It is becoming clearer that voluntary aspects of movement control (preparation, execution and inhibition) are normal in patients who have tics. This current study adds to that growing body of literature. There is a question whether tics are due to inappropriate motor activation (like epileptiform discharges) or due to abnormal motor inhibition. There is evidence that the latter causes tics (see https://doi.org/10.1093/brain/awaa024). I suggest that the authors discuss how their study contributes to how tics are generated vs how they are inhibited.
RESPONSE
We thank the reviewer for this thoughtful comment. We edited paragraphs 3 of Discussion per reviewer’s suggestion.

4.- The authors assess the neurophysiology when stopping might be expected. If I’m not mistaken, this sounds like proactive inhibition (see https://doi.org/10.1038/nrn4038 for a good review). The EEG correlates of proactive inhibition have been extensively studied and some background reading might help the authors frame their finding in the wider context.
RESPONSE
We appreciate this comment and have modified the Introduction and Discussion to reflect this suggestion. The limitation paragraph in Discussion was also edited to address this comment. We did not specifically design the study to examine proactive inhibition. Future study design will have to include variable possibilities of stop trials to behaviorally quantify proactive inhibition.

5.- There are multiple cognitive processes that occur after the car starts moving. Participants prepare to lift their finger as the car approaches the line and (as the authors state) the expectation that a stop signal appears. The current experimental paradigm does not differentiate which of these two processes causes the EEG changes. The study lacks a key control condition – a block of go trials only, which eliminates one of the possibilities above (the expectation that they may need to stop).
RESPONSE
Thank you for this thoughtful comment. We have added this limitation in Discussion.

6.- There are several areas in the manuscript where the authors correlate physiological and clinical measures, and state that there are no significant interactions (without giving specifics of the comparisons made). Please state if the comparisons were corrected for multiple comparisons and if so, how many. This is especially important given that their correlation only just reaches statistical significance (p=0.04).
RESPONSE
We have clarified the Methods section on which variables were included.  We used false discovery rate to correct for multiple comparisons in our exploratory correlation analysis.

7.- Please include a table of the behavioural results including p(inhibit), SSRT, SSD, reaction times, error rates.
RESPONSE
We have included these results in Table 2, including p values.

8.- If p(inhibit) deviates from 0.5, then the authors may want to consider using the integration method rather than the mean method for calculation of the SSRT.
RESPONSE
Our p(inhibit) did not deviate from 0.5. For completeness, we also repeated the analysis using the integration method.  The integration method was performed using an R package called SSRTcalc ( Leontyev, A (2021). SSRTcalc: Easy SSRT Calculation in R. R package version 4.1.0. https://cran.r-project.org/web/packages/SSRTcalc).
The correlation between means/integration method SSRT is highly significant (rho = 0.9, p < 0.0001). Repeat analysis using SSRT_integration_method did not change our results. Since our p(inhibit) did not deviate significantly from 0.5, we chose to keep the means method SSRT for Results.

9.- Preparation of stopping process: I understand why the authors have looked at the time-period between 0 and 300ms (because the stop signal can appear anytime between 300 and 700ms. I wonder if the authors are missing out on significant information about movement preparation by not extending the timepoint beyond 300ms. Indeed, the mean SSD for each participant will be known and instead, the analysis can be re-performed from 0ms to the mean SSD for each participant.
RESPONSE
We thank the reviewer for this interesting suggestion. We repeated our analysis using the mean SSD. The correlation of these EEG power changes and clinical symptoms were not significant. Therefore, our overall findings remained unchanged. We speculate that one of the explanations could be that using the mean SSD could still result in contamination of the EEG power with activity from reactive inhibition. In a single subject, the trial SSD will be shorter than the mean SSD for the subject in about 50% of the stop trials. Since we extracted EEG power after collapsing all stop trials, the period from 0 ms to the mean SSD is likely to include EEG data from some trials in which the stop signal has already been presented, triggering reactive inhibition mechanisms. For this reason, we have kept the analysis up to 300 ms in the manuscript.

Round 2

Reviewer 1 Report
The authors satisfied all my requests

Reviewer 2 Report
The authors addressed my suggested reviews/concerns adequately.